# The acute effect of wearable resistance load and placement upon change of direction performance in soccer players

**Johannes Istvan Rydså**◉, **Roland van den Tillaar**◉*◉

Department of Sport Sciences and Physical Education, Nord University, Levanger, Norway

◉ These authors contributed equally to this work.

* roland.v.tillaar@nord.no

**Data Availability Statement:** All relevant data are within the manuscript and its Supporting Information files.

## Abstract

The aim of the study was to examine the acute effect of different lower limb wearable resistance on placement (shank vs thigh) and various loads (1−5% of body mass) upon change of direction (COD) ability. Twelve male soccer players (age: 23.3 ± 2.5 years; height: 179.2 ± 7.4 cm; body mass: 78.3 ± 7.1 kg) performed a change of direction test with different additional loads fixed on either the shank or thigh. Measurement consisted of total time, 90˚ and 45˚ split times. large effects of the different wearable resistance placement (p<0.05) and load (p<0.001) were found for total and split change of direction time performance. Change of direction times were higher with shank loading compared with thigh loading. It was concluded that lower limb wearable resistance loading with different loads had an acute effect upon change of direction performance in male soccer players. Furthermore, that distal placement (shank vs thigh) with similar body mass load had a larger effect upon COD performance.

## Introduction

Team sports are characterized by frequent episodes of short high-intensity running and longer periods of low-intensity activity [1–3]. These transitions are quite unpredictable and intermittent during a match [4]. Additionally, team sports are characterized by high-intensity movements like sprints, rapid acceleration, deceleration, jumping, blocking, tackling, throwing, kicking and directional changes [5–7]. These movements are important factors to achieve a successful performance in various team sports like rugby union, soccer and Australian football [5, 8, 9]. One of these high-intensity movements is change of direction (COD). In football, a top-class player performs, on average, 726 ± 203 turns during a single match and the equivalent of 609 ± 193 of these turns are conducted in 0˚ to 90˚ in direction to the right or left [10].

There are different training methods to enhance COD performances, where the goal is to improve a greater force output in the athlete's lower limbs [11–13]. The most used training forms are resistance training, ballistic training, plyometric training, assisted training, and traditional sprint training [11–16]. In general, all of these methods have shown a potential positive effect upon COD performance over time [15]. However, more experienced athletes could

**Funding:** The author(s) received no specific funding for this work.

**Competing interests:** The authors have declared that no competing interests exist.

require a greater quantity of specificity, individualization and variations inducted in their strength and conditioning training regime [17]. An optimal transfer is dependent on the specific adaptations to the nature of the training stress, which may only appear with training that mimics to sport competition [12].

One training method in which specificity of training of COD can be preserved is wearable resistance. Wearable resistance training (WRT) is all about using external load on different segments of the body while performing the movements in the various sports, as in CODs [18]. The purpose of WRT is to improve strength and neural activation by a stimulus from the additional load, while simultaneously not being detrimental to the technical embodiment of the specific movement [13]. External loads have been positioned on various places on the body to examine different effects: places such as the head, feet, trunk and arms have been applied in previous studies [19–21]. However, the most common placement area for wearable resistance training is on the trunk and lower limbs [18].

Previous studies on WRT with lower limb loading has shown significant acute effects on walking, running, sprinting and jumping with loads between 0.3 and 8.5% body mass (BM) [18, 22]. With regards to running, lower limb loading placement of ≤1.4% BM appears not to be detrimental to the natural running movements [23, 24]. Furthermore, Simperingham and Cronin [21] reported that trunk loading with 5% BM did not change the sprint performance on 25 m sprint, whereas attachment to the legs with an identical amount of load showed a significant reduction in sprint time on distance ≥10 m. In addition, it is reported that the metabolic response is greater with a more distal attachment than a proximal position [18]. Martin [23] and Field [25] showed that distal lower limb loading had nearly double the effect on oxygen intake compared to proximal loading in endurance running when using wearable loads varying from 0.7% to 5% of BM added. Feser, Macadam [26] reported that shank load placement of 2% of BM during 10 and 50 m sprint resulted in a slightly greater alteration in step kinematics compared to similar loads on the thigh; however, no significant reduction in sprint times was found among unloaded, shank and thigh conditions.

Most studies are performed in a linear movement pattern or vertical direction [18], meaning utilization of wearable resistance during directional changes is yet unexplored. Therefore, the present study will investigate the acute effects of different placements and number of loads attached to the lower limbs (shank and thigh) with team sports athletes on the COD ability. It was hypothesized that wearable resistance would have an acute effect on the COD performance, which makes athletes run acutely slower compared to unresisted CODs. According to the literature, the more distal the placement, the greater the effect on the athlete during a linear movement pattern [23, 25, 26]. This was also hypothesized to affect COD times. Therefore, in this study shank loads were 1, 2, 3% of BM, while thigh loads were 1, 3, 5% BM. Such distribution was hypothesized to make the difference between load placements less, which was reported on metabolic responses during submaximal running [25].

## Materials and methods

### Participants

Twelve healthy and injury-free males (age: 23.3 ± 2.5 years; height: 179.2 ± 7.4 cm; body mass: 78.3 ± 7.1 kg) participated in the study. The subjects were active football players ranging from the second to fifth National Division. All the subjects had experience with some sort of COD tests, but not with wearable resistance training. The study was approved by the Norwegian Centre for Research Data and performed according to the Declaration of Helsinki. All the participants were fully informed of the nature of the study before providing their written consent to participate. The experiment was conducted in November–December when the season's

competition had just ended. Additionally, subjects were informed to avoid strenuous training for 24 hours, consumption of alcohol for at least 12 hours, and consumption of a heavy meal less than 2 hours before each session.

## Procedures

In order to compare the effects of different loads and placement of wearable resistance on lower limbs on COD performance, a repeated measurement design was applied in which the subjects performed a COD test with three different loads and two different load placements (shank and thigh). The independent variables were six different conditions used when performing the COD test, and the dependent variables were total time and split time of the COD test. To avoid any form of learning effects, a familiarization day was applied, and the testing was conducted with a randomized cross-over design with enough time to clear out any chronic effects.

The testing was conducted on two different occasions, consisting of one familiarization day and one testing day, which were conducted with 2–7 days between each other. All the sessions started with a standardized warm-up protocol as specified by van den Tillaar and von Heimburg [27], which consisted of a total 10 min with 1 of 7 different dynamic stretch exercises that were performed in the recovery period of 60 s between the 8 x 40 m runs, and the runs were performed at self-estimated intensity, starting from 60% of maximal sprinting velocity and then increasing by 5% until reaching 95% [27]. After 2 min rest, the subjects performed two submaximal runs in the COD test with 2 min recovery in between. This was followed by performing the test with maximal effort. Each subject had two attempts in each condition. The order of the loads and placements were randomised for each subject to avoid a fatigue or learning effect.

The familiarization session consisted of collecting basic anthropometric data such as height and weight and making subjects familiar with equipment and procedure. During this session subjects performed two trials in the COD test with each load and both placements. Hence, 2 x unresisted runs, 2 x 1, 2, and 3% BM with shank placement and 2 x 1, 3, and 5% BM with thigh placement. After each run, a rest period of 2–3 min was conducted to avoid fatigue [28, 29].

The test day consisted of two runs on each of the seven different load conditions and used identical recovery time as in the familiarization session. On the familiarization day, all subjects started with unresisted runs, and then loads were fixed in a random order, while on the test day all runs were randomized. Every subject performed two sprints in each condition before changing to the next condition in which new loads were fixed/removed during the recovery periods. All the testing and warm-ups for all the sessions were conducted in an indoor hall and the subjects wore their own indoor soccer/jogging shoes.

The subjects started in a standing position 0.3 m behind a pair of photocells (Browser Timing Systems, Draper, UT, USA). The COD test contained a total distance of 25 m with 5 m between each turn with the first two turns of 90˚ (one right and one left turn) followed by two turns of 45˚. Each cone was placed 20 cm with an angle of 45˚ from every turning point (Fig 1). Total sprint times after four turns together with the split sprint times after the two 90˚ turns and the two 45˚ turns were measured with three pairs of photocells (Browser Timing Systems, Draper, UT, USA). The first pair had a height of 0.3 m while the last two pairs of photocells were placed at a height of 0.7 m (Fig 1).

The subjects wore a pair of leg sleeves and compression shorts (Lila™, Sportboleh Sdh Bhd, Kuala Lumpur, Malaysia) with different loads attached to them. The load was regulated using fusiform shaped loads of 50, 100, 200, and 300 g, which could be attached to the garments by a Velcro backing. All the loads were estimated to the nearest 50 g on every run and 300 g were

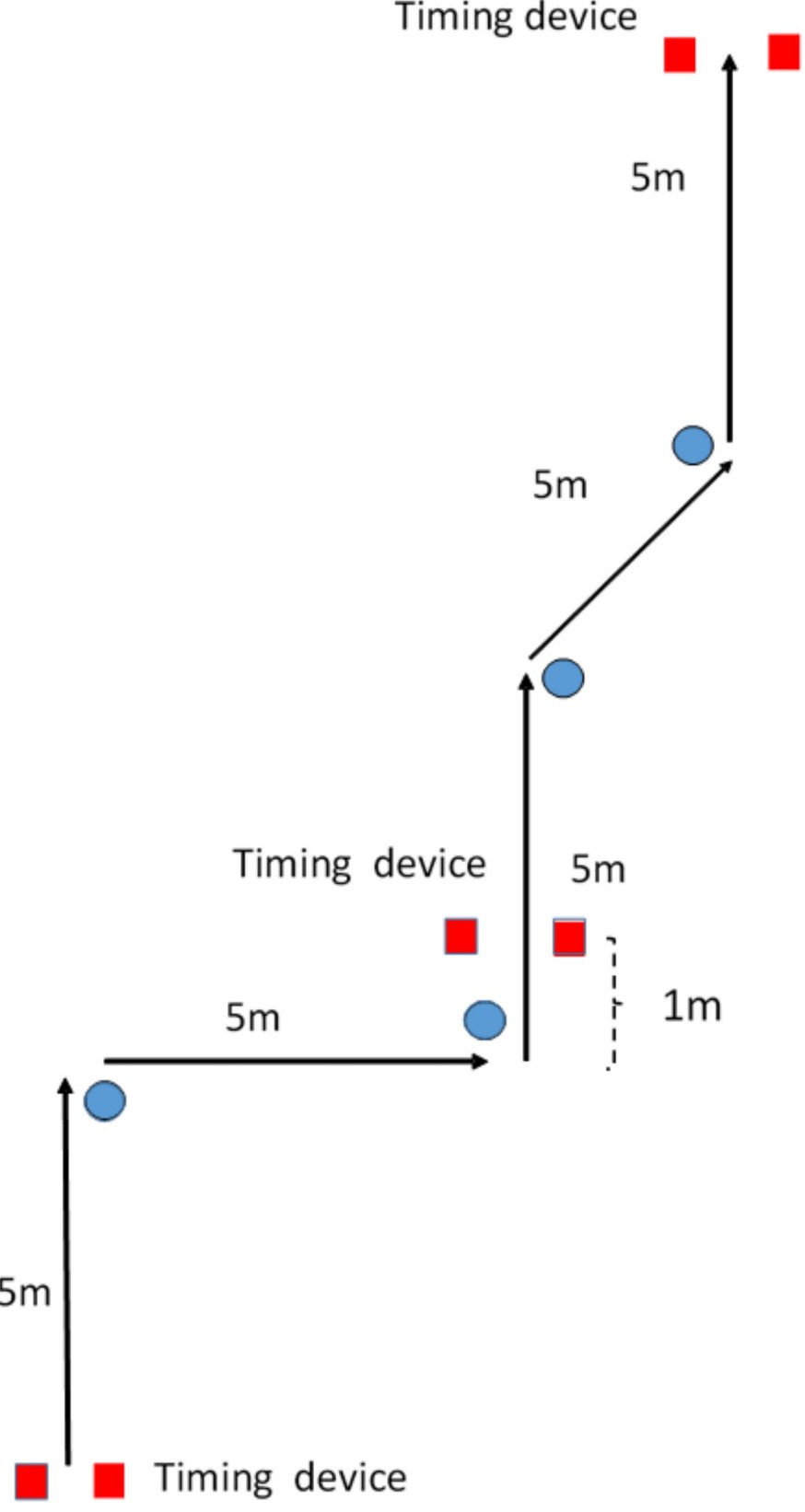

**Fig 1. Change of Direction (COD) test with two 90° followed by two 45° turns with timing after the two 90° and 45° turns.**

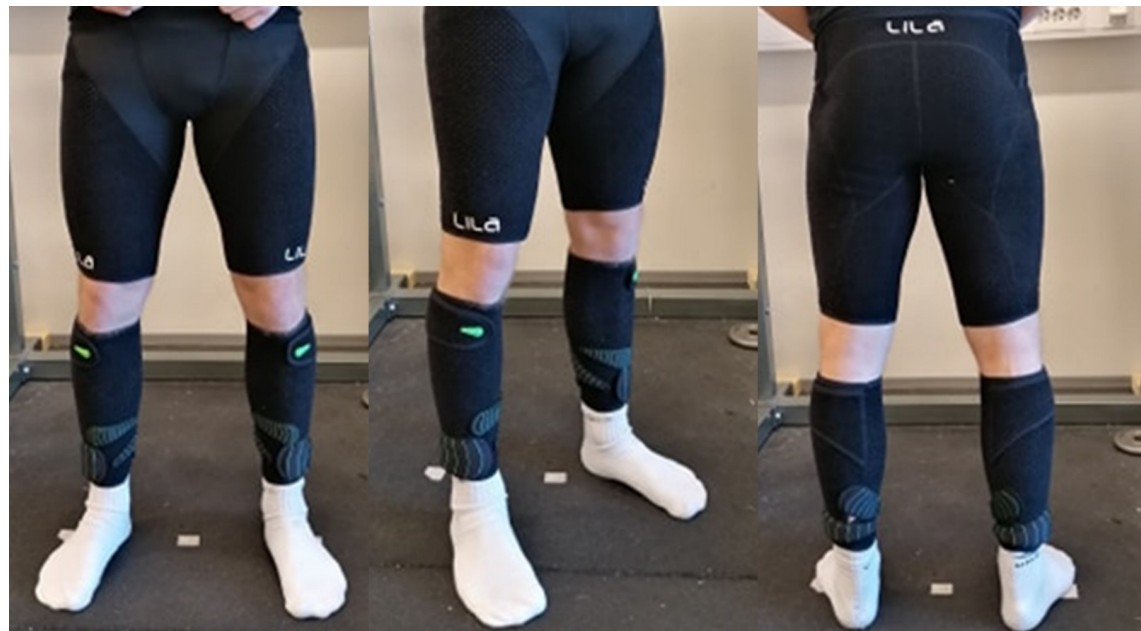

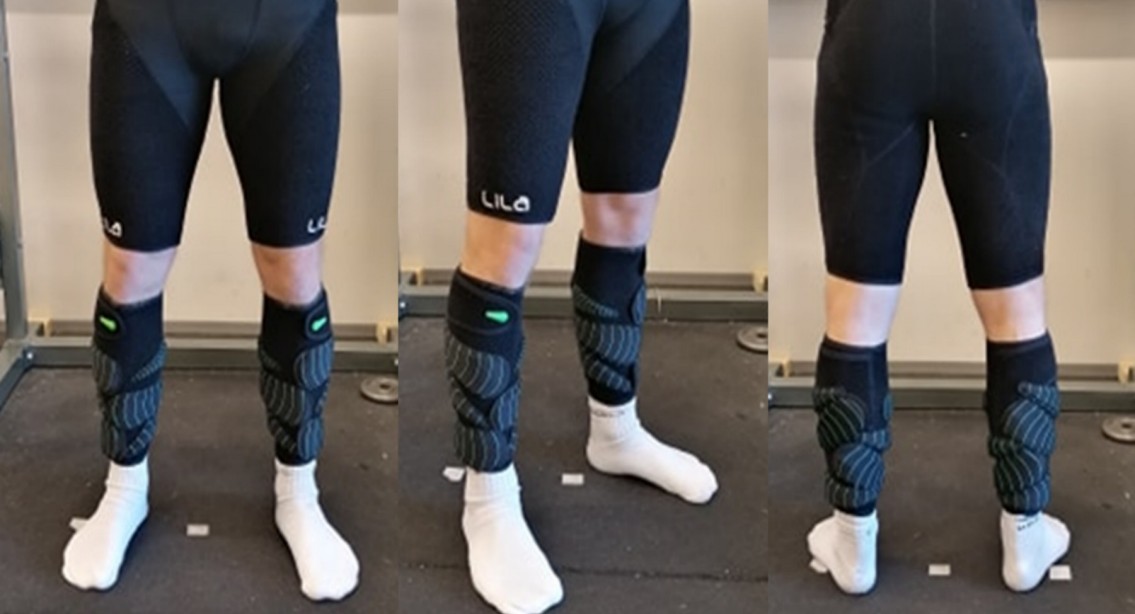

**Fig 2. Illustration of 1 and 3% BM resistance with shank condition for a 70 kg football player.**

only used on the thigh condition. Placement of the loads was in accordance with the loading scheme protocol of Field [25], but with some practical modifications. Loads were placed from most distal to proximal on the leg. The first load was attached horizontally, laterally and at the most distal point with the heaviest part placed anteriorly on the leg, while the next load was placed the opposite way (medial) with the heaviest part posterior on the leg. Additionally, the thigh condition loads were placed more laterally to avoid affecting the natural running gait during the heaviest loading (Figs 2 and 3).

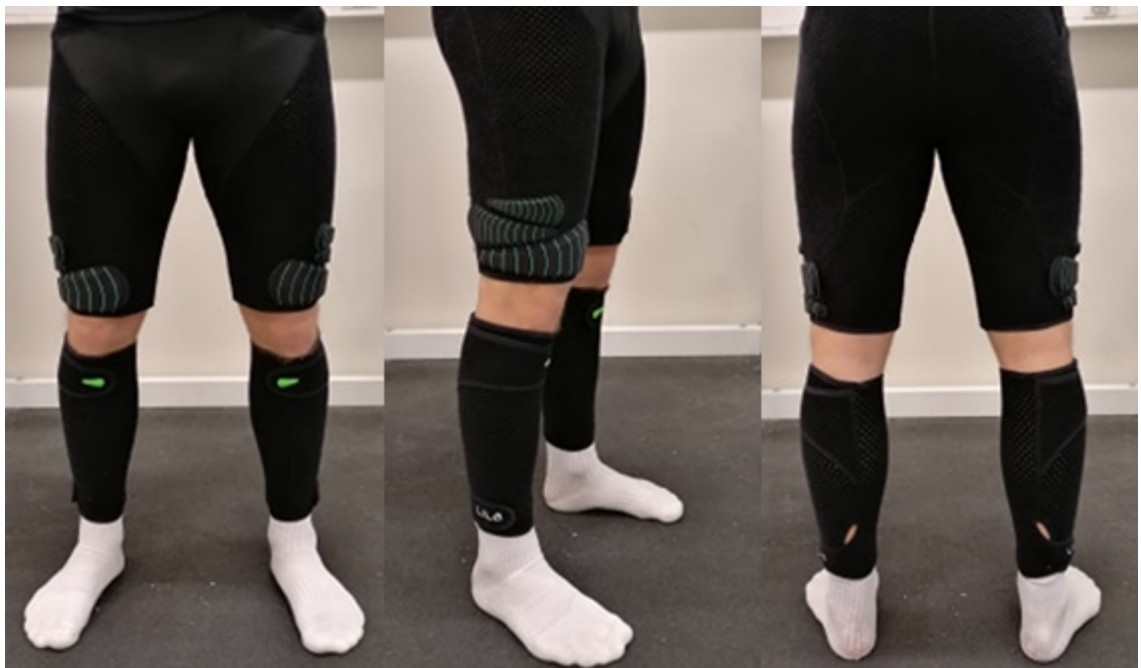

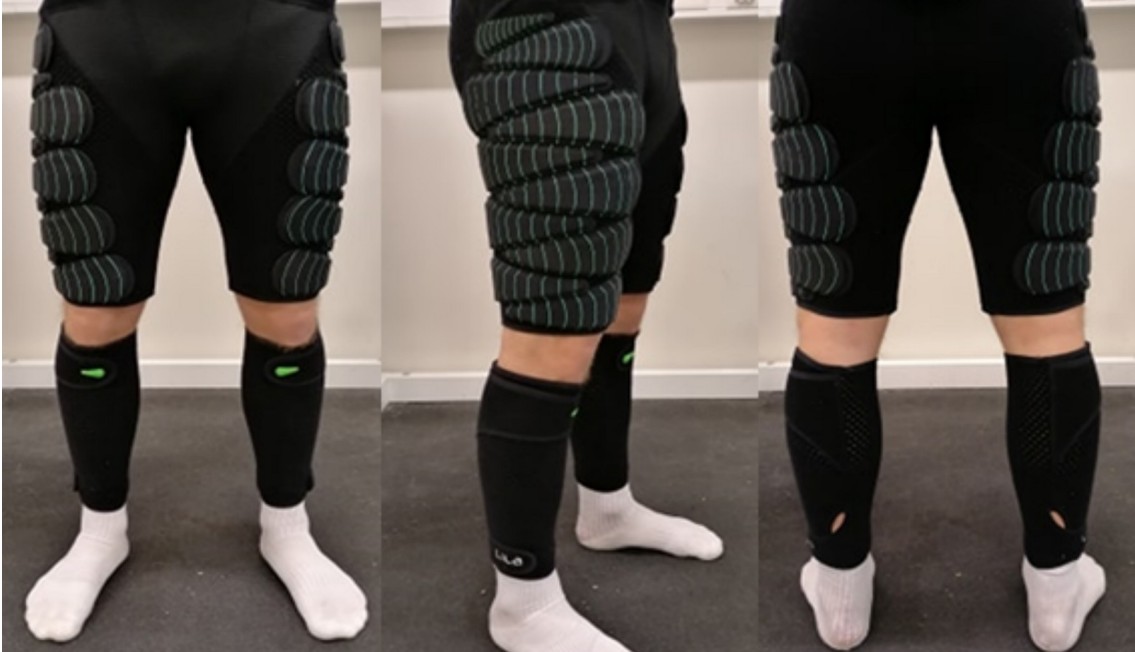

**Fig 3. Illustration of 1 and 5% BM resistance with thigh condition for a 70 kg football player.**

## Statistical analysis

Descriptive statistics were presented as means and standard deviations. Data were checked for normal distribution using the Shapiro–Wilk test. A 2 (placement: shank vs thigh) x 4 (load) analysis of variance (ANOVA) with repeated measures was used to compare the effects of different wearable resistance conditions upon total and split times (90˚ and 45˚). After significant differences were located on sprint times and split times, post hoc comparisons with Holm-

Bonferroni corrections was applied to determine exactly where the differences occurred. If p-values for sphericity (Mauchly's test) assumptions were violated, corrections with Greenhouse −Geisser were reported. The level of significance was set at p < 0.05. Analysis was performed with SPSS Statistics for Windows, version 25.0 (IBM Corp., Armonk, NY, USA). Effect size (ES) was evaluated with according to Cohens d, and interpretations of the magnitude were as follows: 0–0.2 = trivial, 0.2–0.6 = small, 0.6–1.2 = moderate, 1.2–2.0 = large, and >2 = very large [30].

## Results

A large effect of the different wearable resistance placement was found for total and split COD times (F ≥ 5.4; P ≤ 0.040; 0. d ≥ 1.4). In addition, a very large effect was found for load in total and split times (F ≥ 11.4; P < 0.001; d ≥ 2.0) with only trivial to moderate interaction effects (F ≤ 1.9; P≥ 0.155; 0.2 ≤ d ≤ 0.8). Post hoc comparison revealed that COD times were higher with shank loading compared with thigh loading. However, when compared pairwise, only the 90˚ split times with 1% load were longer with placement on the shank compared with on the thigh (p = 0.032, Fig 4). Furthermore, comparing the same 3% load with different placement showed moderately higher COD times when placed on the shank compared to the thigh on both total time (p = 0.004, d = 1.0) and 90˚ split times (p = 0.003, d = 0.9), while 45˚ split times (p = 0.09) showed only a small difference (d = 0.4) between the conditions.

Moreover, the unresisted COD times were shorter in total time and in the 45˚ split times compared to the loading conditions, while with the 90˚ split times they increased with 2% shank and 3% thigh loading. Furthermore, did total COD time increase between 1% with 3% shank and 1 to 3% thigh loading (large effect). 90˚ split times also increase between 1 and 3% thigh loading and between 2 and 3% shank loading (moderate effect). 45˚ split times only increased between 1 and 5% thigh loading (large effect), while only a moderate effect was visible between 1 and 2 shank and 1 and 3 thigh loadings (Fig 4).

## Discussion

The purpose of this study was to investigate the acute effect of different placement and loads of wearable resistance attached to the lower limbs on the change of direction ability in football players. The main findings were that wearable resistance placed on the lower limbs with loads of 1−5% of BM increased total and split times in a COD performance and that this effect was larger with shank loading compared with thigh loading.

The acute increase in time with lower limb loading on the COD test is comparable with the findings of Simperingham and Cronin [21] and Simperingham, Cronin [31], who reported an increase in total sprint time (3.3 and 2.0%, respectively) during respectively 25 m non-motorized treadmill sprinting and 20 m over-ground sprinting with a loading placed of 5% BM on both shank and thigh (whole leg). This was similar to the total distance of the COD track. In contrast to the present study, the use of a lower loading of 3% of BM placed on the whole leg showed no differences in sprint time compared with unresisted conditions during a 20 m sprint [31, 32]. Furthermore, Feser, Macadam [26] reported no differences between using shank or thigh placement of 2% of BM loading on 10 m sprint time compared to unresisted sprinting, indicating that wearable resistance seems to have a larger impact upon COD movements than on straight-line sprinting. This is also explainable by the fact that due to the change of direction the limbs must decelerate and re-accelerate more in several directions, which would cost more energy and coordination. Furthermore, the influence of small loads of wearable resistance (1%) seems to have more effect when the athlete is already at speed. In the present study, a noticeable effect with 1% loading was found in the second part of the COD track

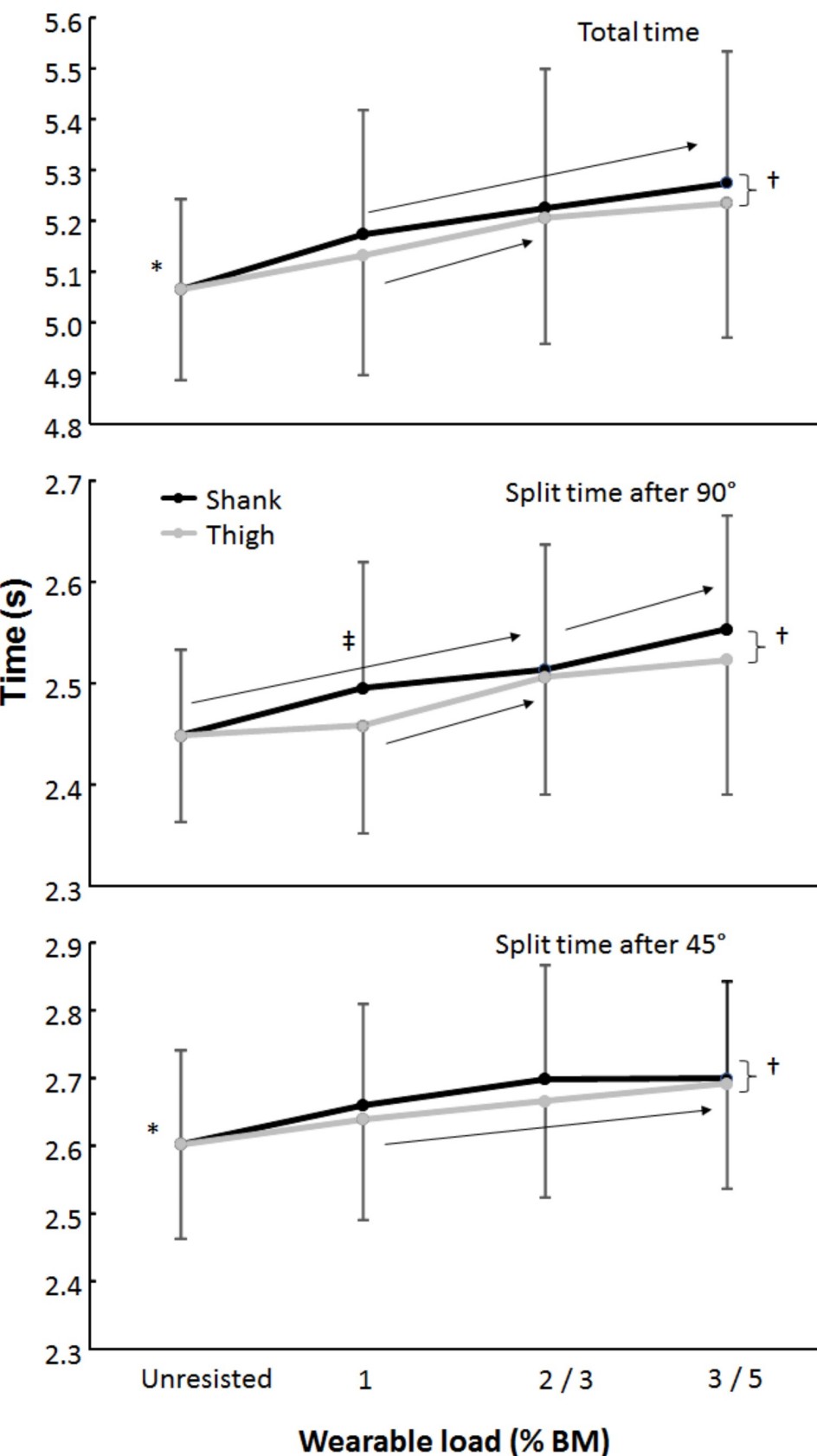

**Fig 4. Total and split times after 90˚ and 45˚ CODs (mean ± SD) for unresisted and with 1−3% body mass load on the shank and 1, 3, 5% body mass load on the thigh.** † indicates a difference in time (p < 0.05) between the two conditions (shank vs thigh). * indicates a difference in time (p < 0.05) with all loaded conditions. ‡ indicates a difference in time (p < 0.05) between shank and thigh condition with this load. → indicates a difference between these two loads for this condition and all those to the right of it on a p < 0.05 level.

(45˚ split times), which was in accordance with 2.4−3% of BM loading on whole leg during the 10 m to 20 m in a sprint [32, 33].

The effect of lower wearable resistance was larger with shank loading compared to thigh loading, which was consistent with previous studies in terms of velocity, kinematics, and metabolic response measurements to distinguish between shank and thigh loading during running and sprinting [25, 26]. Furthermore, shank placement provides a greater effect stimulus compared to thigh placement when using identical loads of 1 and 3% [25, 26], which is the result of greater inertia due to a more distal loading [18]. At 1% loading only a trivial difference in the 90˚ split times was found (+1.5%). This can be explained by the fact that it costs more time to initiate load and accelerate it when placed more distally on the limb due to moment of inertia, as Feser, Macadam [26] reported by lower step frequency (-2.1%) with shank placement regarding the acceleration phase of the sprint. Furthermore, with 90˚ turns the participant has to decelerate more each time and re-accelerate again than with 45˚ turns [34], which is also influenced more by distal loading as shown by the increased times with the increased loads on the shank (2 to 3% load). When the participant is at full speed, the difference in load placement is much less and therefore no differences were found for the 45˚ split times. This was also visible with the 3% load in which the total time and 90˚ split times were higher with shank load placement vs thigh loading and no differences were found with the 45˚ split times (Fig 4).

In addition, the 45˚ split times only increased between 1 and 5% on thigh loading, which may be due to the fact that shank loading of 1% BM was already causing a lot of overload to the athletes. In this case, the latter from 1 to 3% shank loading is not as steep compared to the 1 to 5% thigh loading, but from unloaded to 1% BM the process is considerably greater. Similar findings were also reported in acute oxygen consumption during submaximal running between the two placements [25].

Some limitations of the present study were that no joint and step kinematic or kinetic and muscle activity measurements were performed that could give more insights into what exactly changes–e.g. shorter steps, lower knee flexion or more proximal muscle use [18, 22, 26, 32] – with wearable resistance during CODs. In addition, only CODs with 45˚ and 90˚ turns were performed, which are mainly used in football [10], while it is not certain what the influence of wearable resistance on lower limbs is in turns of more than 90˚ change of direction. Another limitation was that the subjects were male football players, which leaves the knowledge of the effect of wearable resistance with similar lower limb loads in female athletes unknown. Furthermore, only acute effects of load and placement upon COD performance were examined and so this does not necessarily explain the longitudinal effects of wearable resistance upon COD performance. The lack of knowledge should be considered in terms of future studies, in which kinematic, kinetic and muscle activation measurement should be included and longitudinal effects of wearable resistance on the lower limbs on COD performance in team sport players should be conducted.

## Conclusion

It was concluded that lower limb wearable resistance loading with different loads had an acute effect on change of direction performance in male football players. Furthermore, it was demonstrated that distal placement (shank vs thigh) with similar BM load had a larger effect on

COD performance, particularly in turns with 90˚ compared with 45˚, which is probably due to the increased moment of inertia during accelerating and decelerating the limbs during the turns. Therefore, shank loading during training could have a larger effect on performance with even less loading to induce some adaptation. However, training interventions have to be conducted to investigate whether these loadings could have a positive longitudinal effect upon COD performance.

## Supporting information

**S1 File. COD times data.**
(XLSX)

## Author Contributions

**Conceptualization:** Johannes Istvan Rydså, Roland van den Tillaar.

**Formal analysis:** Johannes Istvan Rydså, Roland van den Tillaar.

**Funding acquisition:** Johannes Istvan Rydså.

**Investigation:** Johannes Istvan Rydså, Roland van den Tillaar.

**Methodology:** Johannes Istvan Rydså, Roland van den Tillaar.

**Supervision:** Roland van den Tillaar.

**Writing – original draft:** Johannes Istvan Rydså, Roland van den Tillaar.

**Writing – review & editing:** Roland van den Tillaar.

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
