## [Decision Letter · Decision Letter 0]

14 Oct 2020

PONE-D-20-09633

The acute effect of wearable resistance load and placement upon change of direction performance in soccer players.

PLOS ONE

Dear Dr. van den Tillaar,

Thank you for submitting your manuscript to PLOS ONE. After careful consideration, we feel that it has merit but does not fully meet PLOS ONE’s publication criteria as it currently stands. Therefore, we invite you to submit a revised version of the manuscript that addresses the points raised by the second reviewer.

We look forward to receiving your revised manuscript.

Kind regards,

Professor Dominic Micklewright, PhD CPsychol PFHEA FBASES FACSM

Academic Editor

PLOS ONE

Journal Requirements:

Reviewers' comments:

Reviewer's Responses to Questions

**Comments to the Author**

1. Is the manuscript technically sound, and do the data support the conclusions?

Reviewer #1: Yes

Reviewer #2: Yes

2. Has the statistical analysis been performed appropriately and rigorously? 

Reviewer #1: Yes

Reviewer #2: Yes

3. Have the authors made all data underlying the findings in their manuscript fully available?

Reviewer #1: Yes

Reviewer #2: Yes

4. Is the manuscript presented in an intelligible fashion and written in standard English?

Reviewer #1: Yes

Reviewer #2: Yes

5. Review Comments to the Author

Reviewer #1: Well planned and conducted study into a growing area of research: WR. The COD aspect is a new area and adds to the sprint studies done to date.

Well planned and conducted study into a growing area of research: WR. The COD aspect is a new area and adds to the sprint studies done to date.

Well planned and conducted study into a growing area of research: WR. The COD aspect is a new area and adds to the sprint studies done to date.

Well planned and conducted study into a growing area of research: WR. The COD aspect is a new area and adds to the sprint studies done to date.

Well planned and conducted study into a growing area of research: WR. The COD aspect is a new area and adds to the sprint studies done to date.

Well planned and conducted study into a growing area of research: WR. The COD aspect is a new area and adds to the sprint studies done to date.

Well planned and conducted study into a growing area of research: WR. The COD aspect is a new area and adds to the sprint studies done to date.

Reviewer #2: Thank you for the opportunity to review this article.

This study aimed to examine the acute effect of different lower limb wearable resistance on placement (shank vs. thigh) and various loads (1−5% of body mass) upon change of direction (COD) ability.

Comments:

Introduction:

Page 9 line35-36: please add a reference.

Line 40: football should be soccer

Page 12 line 110: please put a space between of and one.

Line 116 please change two minutes by 2 min, like the next sentence.

Statistical analyses:

It should be better to convert partial eta squared to Cohens’d.

Results:

It is time to stop using the term “statistically significant” entirely. Nor should variants such as “significantly different,” “p < 0.05,” and “nonsignificant” survive, whether expressed in words, by asterisks in a table, or in some other way.’

https://www.tandfonline.com/doi/pdf/10.1080/00031305.2019.1583913

6. PLOS authors have the option to publish the peer review history of their article (what does this mean?). If published, this will include your full peer review and any attached files.

Reviewer #1: No

Reviewer #2: **Yes: **Yassine Negra

---

## [Author Response · Author response to Decision Letter 0]

17 Oct 2020

Reviewer #1: Well planned and conducted study into a growing area of research: WR. The COD aspect is a new area and adds to the sprint studies done to date.

Reviewer #2: Thank you for the opportunity to review this article.

This study aimed to examine the acute effect of different lower limb wearable resistance on placement (shank vs. thigh) and various loads (1−5% of body mass) upon change of direction (COD) ability.

Comments:

Introduction:

Page 9 line35-36: please add a reference.

We have added 3 references.

Line 40: football should be soccer

Changed now.

Page 12 line 110: please put a space between of and one.

Changed now.

Line 116 please change two minutes by 2 min, like the next sentence.

Changed it now.

Statistical analyses:

It should be better to convert partial eta squared to Cohens’d.

We have converted the partial eta squared to Cohen’s d

Results:

It is time to stop using the term “statistically significant” entirely. Nor should variants such as “significantly different,” “p < 0.05,” and “nonsignificant” survive, whether expressed in words, by asterisks in a table, or in some other way.’

https://www.tandfonline.com/doi/pdf/10.1080/00031305.2019.1583913

We have read the article and changed the significant difference in small, moderate and large effects as suggested by Hopkins et al. (2009). We still use asterisk in figure since it showed that these differences were larger than p < 0.05, but also had a large effect, which again is written in the text instead of significance. We have taken all significance wording away from the results and discussion parts.

---

## [Editor Report · Decision Letter 1]

4 Nov 2020

The acute effect of wearable resistance load and placement upon change of direction performance in soccer players.

PONE-D-20-09633R1

Dear Dr. van den Tillaar,

We’re pleased to inform you that your manuscript has been judged scientifically suitable for publication and will be formally accepted for publication once it meets all outstanding technical requirements.

Kind regards,

Dominic Micklewright, PhD CPsychol PFHEA FBASES FACSM

Academic Editor

PLOS ONE

---

## [Editor Report · Acceptance letter]

9 Nov 2020

PONE-D-20-09633R1 

The acute effect of wearable resistance load and placement upon change of direction performance in soccer players. 

Dear Dr. van den Tillaar:

I'm pleased to inform you that your manuscript has been deemed suitable for publication in PLOS ONE. Congratulations! Your manuscript is now with our production department. 

Kind regards, 

on behalf of

Professor Dominic Micklewright 

Academic Editor

PLOS ONE